# The Fungal Endophyte *Penicillium olsonii* ML37 Reduces Fusarium Head Blight by Local Induced Resistance in Wheat Spikes

**DOI:** 10.3390/jof8040345

**Published:** 2022-03-25

**Authors:** Edward C. Rojas, Birgit Jensen, Hans J. L. Jørgensen, Meike A. C. Latz, Pilar Esteban, David B. Collinge

**Affiliations:** 1Section for Microbial Ecology and Biotechnology, Department of Plant and Environmental Sciences, University of Copenhagen, Thorvaldsensvej 40, 1871 Copenhagen, Denmark; bje@plen.ku.dk (B.J.); pitu_es@hotmail.com (P.E.); 2Copenhagen Plant Science Centre, University of Copenhagen, Thorvaldsensvej 40, 1871 Copenhagen, Denmark; hjo@plen.ku.dk (H.J.L.J.); meike.latz@scilifelab.se (M.A.C.L.); 3Chr Hansen A/S, Højbakkegård Alle 30, 2630 Tåstrup, Denmark; 4Section for Plant and Soil Science, Department of Plant and Environmental Sciences, University of Copenhagen, Thorvaldsensvej 40, 1871 Copenhagen, Denmark; 5SciLifeLab, KTH Royal Institute of Technology, 171 65 Solna, Sweden; 6Department of Agricultural, Food and Agro-Environmental Sciences, University of Pisa, Via del Borghetto 80, 56124 Pisa, Italy

**Keywords:** asperphenamate, biological control, culmorin, endophyte, Fusarium head blight, induced resistance, *Penicillium olsonii*, RNA-seq, specialised metabolite, mycotoxin

## Abstract

The fungal endophyte *Penicillium olsonii* ML37 is a biocontrol agent of Fusarium head blight in wheat (caused by *Fusarium graminearum*), which has shown a limited direct inhibition of fungal growth in vitro. We used RNA-seq and LC-MS/MS analyses to elucidate metabolic interactions of the three-way system Penicillium–wheat–Fusarium in greenhouse experiments. We demonstrated that *P. olsonii* ML37 colonises wheat spikes and transiently activates plant defence mechanisms, as pretreated spikes show a faster and stronger expression of the defence metabolism during the first 24 h after pathogen inoculation. This effect was transient and the expression of the same genes was lower in the pathogen-infected spikes than in those infected by *P. olsonii* alone. This response to the endophyte includes the transcriptional activation of several WRKY transcription factors. This early activation is associated with a reduction in FHB symptoms and significantly lower levels of the *F. graminearum* metabolites 15-acetyl-DON and culmorin. An increase in the Penicillium-associated metabolite asperphanamate confirms colonisation by the endophyte. Our results suggest that the mode of action used by *P. olsonii* ML37 is via a local defence activation in wheat spikes, and that this fungus has potential as a novel biological alternative in wheat disease control.

## 1. Introduction

Fusarium head blight (FHB) limits wheat productivity around the world [1]. Infection occurs during anthesis, and the disease affects the development of the kernels directly. In addition to reducing yield, the disease affects grain quality due to contamination with mycotoxins such as fumonisins, trichothecenes and zearalenone [2]. These mycotoxins pose a health risk when consumed [3]. The management of FHB remains a challenge due to limited genetic resistance and narrow windows for fungicide applications [4].

*Fusarium graminearum* and *F. culmorum* in particular pose problems. Fungal spores germinate and the pathogens colonise the plant’s surface using runner hyphae within 12 h after initial contact with the spikelets. Infection occurs between 24 and 48 h after pathogen spore arrival. Hyphae branch and aggregate, forming appressoria and infection cushions [5]. The fungi initially exhibit a biotrophic phase, where they spread through the intercellular spaces asymptomatically. Nevertheless, molecular interactions take place during this phase. Both mycotoxin production and plant defence mechanisms are highly active during this stage [6]. A later necrotrophic phase is most likely induced by nutrient depletion within plant tissues [7]. During this later phase, fungal hyphae degrade colonised tissues, causing bleached spike symptoms [6]. Subsequently, the pathogens produce aerial sporodochia on the necrotized tissue. 

Wheat plants detect *Fusarium* spp. infection during the early stages of the interaction. The high expression of transcription factors associated with PAMP recognition and signal transduction proteins, such as WRKY and bZIP, takes place during the first phase of infection [8]. In general, this is associated with an activation of the salicylic acid (SA) pathway during the biotrophic phase and the jasmonate-ethylene pathway during the necrotrophic phase, although the roles of these hormones are still under debate [9,10].

Pathogenesis-related genes such as *PR-1*, *PR-2* (β-1,3-glucanase), *PR-3* (chitinase), *PR-4*, *PR-5* (thaumatin-like protein) and oxidative stress-related enzymes are highly expressed during the first 12 h after inoculation and peak between 36 and 48 h [11,12,13]. Most of these responses have been associated with QTLs in partially resistant cultivars [14]. However, complete genetic resistance in wheat cultivars remains elusive and partial resistance is polygenic and scattered across the genome [15]. Interestingly, transgenic approaches for conferring resistance against *Fusarium* are promising, if not accepted globally [16,17,18,19].

Biological control is receiving increased attention as an alternative approach for disease control because it can provide new modes of action whilst enjoying high acceptance among consumers [20,21]. 

Several studies have shown different levels of FHB biocontrol by living organisms [22]. Bacteria from the genera *Bacillus* and *Pseudomonas* have been shown to reduce levels of FHB in greenhouse studies when vegetative cells or spores were sprayed at anthesis [23,24,25,26]. Among fungal candidates, *Clonostachys rosea*, *Cryptococcus* sp. and *Sarocladium zeae* have also been shown to suppress FHB in greenhouse studies when spores were applied at heading [27,28,29]. Most of these studies using microbes have shown an ability to control the disease, but little is known about their mode of action. 

Recently, we demonstrated how the fungal endophyte *Penicillium olsonii* ML37, isolated from wheat leaves, reduced the severity of Septoria tritici blotch [30] and Fusarium head blight [31]. Thus, Rojas et al. [31] found that ML37 reduced FHB symptoms and pathogen biomass in wheat spikes when inoculated before anthesis in greenhouse assays. Interestingly, ML37 did not inhibit *F. graminearum* growth in vitro and required a lag period before pathogen inoculation to provide a significant biocontrol effect. This suggests that the disease-reducing effect of ML37 is mediated by the plant after inoculation with the pathogen.

The aim of this study is to investigate the mode of action of the biocontrol mediated by *P. olsonii* ML37 against FHB and, specifically, whether the activation of host defence responses is involved. We perform a transcriptional analysis using the RNA-seq analysis on endophyte-treated and nontreated spikes. Likewise, we evaluate the transcriptomic profile of treated spikes after *F. graminearum* infection and analyse the content of mycotoxins and other specialised metabolites in the treated spikes. Our findings suggest that biocontrol mediated by ML37 is at least in part triggered by the activation of inherent plant resistance.

## 2. Materials and Methods

### 2.1. Plant and Fungal Material

Spring wheat cv. Diskett (SW 45456-Lantmännen) was grown to heading stage in a greenhouse. This cultivar is moderately susceptible to Fusarium head blight. The endophyte *Penicillium olsonii* ML37 was isolated in our previous study [30]. Agar plugs were cultured on potato dextrose agar (PDA) (Difco™, BD, Franklin Lakes, NJ, USA) plates for 7 days. Spores were harvested by flooding the plate using autoclaved MilliQ water + Tween 20 (0.05% *v*/*v*). The spore suspension was filtered using Miracloth^®^ (Merck, Darmstadt, Germany), centrifuged and resuspended. The *P. olsonii* spore concentration was adjusted to 10^7^ spores per ml using MilliQ water (Merck KGaA, Darmstadt, Germany) + Tween 20 (Thermo Scientific™, Waltham, Massachusetts, USA) (0.05% *v*/*v*).

*F. graminearum* strain WC-091-7 was kindly provided by Dr. Lise N. Jørgensen, Aarhus University. Agar plugs were added to 20 mL RA liquid medium (Imholte and Schramm, 1968) and cultured for 10 days. Macroconidia were collected by filtering the culture through autoclaved Miracloth^®^ (Merck). Spores were centrifuged and resuspended and the *F. graminearum* spore concentration was adjusted to 10^5^ macroconidia per ml using autoclaved MilliQ water + Tween 20 (0.05% *v*/*v*).

In order to identify the transcriptomic and enzymatic changes on wheat spikes induced by *P. olsonii* ML37 alone (before anthesis) and in the presence of *Fusarium* infection (after anthesis), four separate experiments were performed, namely, experiments 1, 1.1, 2 and 2.2.

**Experiment 1**. *P. olsonii**ML37 transcriptomic effect on heading-stage wheat spikes before anthesis*. Heading-stage wheat spikes (BBCH 58) were sprayed with either a *P. olsonii* ML37 spore suspension or autoclaved MilliQ water + Tween as a mock treatment using a 100 mL spray bottle. Approximately 2 mL was sprayed onto each spike until run-off. Treatments were applied at late afternoon to prevent exposure of the treated spikes to direct sunlight. The spikes were left uncovered and harvested at 48 h after inoculation. Samples were immediately snap-frozen in liquid nitrogen and stored at −80 °C until RNA extraction. A total of three replications (spikes) per treatment was harvested.

**Experiment 1.1**. *β-1,3-glucanase activity of P. olsonii ML37-treated heading-stage wheat spikes before anthesis.* Experiment 1 was repeated to evaluate the β-1,3-glucanase activity at three different time points (24, 48 and 72 h), following the protocol of Shetty et al. [32]. Four replications of each condition were used. Total protein was extracted from 200 mg ground tissue using 250 µL 0.05 M sodium acetate buffer (pH 5.2). Specific β-1,3-glucanase activity was calculated based on the absorbance at 540 nm using the absorbance in an Eon™ Microplate Spectrophotometer reader (Winooski, VT, USA). Specific enzyme activity was calculated as μmol released product/min for every mg of initial protein. Data were analysed using a generalized linear model. Comparisons between treatments at each time point were considered significant at *p* < 0.05.

**Experiment 2.** *P. olsonii ML37 transcriptomic effect on wheat spikes during Fusarium graminearum infection after anthesis.* Heading-stage wheat spikes were inoculated with *P. olsonii* ML37 spores or MilliQ water + Tween as described above. Between three and five days later, as spikes reached mid-anthesis stage (BBCH 65), all the spikes were inoculated with a *F. graminearum* WC-091-7 macroconidial suspension (10^5^ spores/mL) by spraying approximately 2 mL until run-off. Treated spikes were covered with a previously mist-treated transparent plastic bag and sealed using adhesive tape around the straw to increase humidity around the spike. Plastic bags were maintained until harvest of each time point. Spikes were harvested at 24 and 72 h after inoculation with the pathogen. In total, four replications were harvested for each condition and time point. Spikes were snap frozen in liquid nitrogen immediately after harvest and stored at −80 °C until RNA extraction. 

The remaining spikes were inspected for FHB severity at 7 days after inoculation. Severity was calculated as the number of spikelets exhibiting visible FHB symptoms out of the total number of spikelets in each spike. Statistical analysis of FHB severity was performed using a generalized linear model specifying a binomial distribution. All statistical comparisons were performed against FHB controls and were considered significant when *p* ≤ 0.05.

**Experiment 2.1.** *β-1,3-glucanase activity of P. olsonii ML37-treated wheat spikes during* Fusarium graminearum *infection*. Experiment 2 was repeated to evaluate the β-1,3-glucanase activity at three different time points (24, 48 and 72 h). Four replications of each condition and time point were used. Samples were analysed as described above. 

The remaining spikes were harvested 7 days after pathogen inoculation and stored at −20 °C for multi-mycotoxin and specialised (secondary) metabolite analysis.

### 2.2. RNA Extraction and Illumina Sequencing

Harvested spikes from experiments 1 and 2 were used for RNA extraction. Six spikelets with rachis were dissected from the middle of each treated spike and ground up in liquid nitrogen using a pestle and mortar. Approximately 100 mg of ground tissue was used for RNA extraction. RNA was extracted using the RNeasy Plant Mini Kit (Qiagen, Hilden, Germany). RNA samples were treated with a DNAse treatment using the DNA-free™ DNA removal Kit (Thermo Fisher Scientific, Waltham, MA, USA). RNA quantity and quality were assessed using a NanoDrop 2000 Spectrophotometer (Thermo Scientific™, Waltham, MA, USA) and visualised on a 2% agarose gel. RNA integrity was verified using an Agilent 2100 Bioanalyzer (Thermo Scientific™). In total, 22 samples were used for library preparation and sequencing (6 for experiment 1 and 16 for experiment 2).

Library preparation and Illumina high-throughput sequencing was performed by the NovoGene Sequencing company (Hong Kong, China) using the HiSeq paired-end platform (Illumina, San Diego, CA, USA). Sequencing depth was at least 50 million paired-end reads per sample. Sequence quality and error rate was verified using Q_phred_ scores. 

### 2.3. Bioinformatic Analysis

Raw data were filtered as follows: after demultiplexing, reads containing adapters, reads with Q-score ≤ 5 and reads containing more than 10% N calls were removed. Pseudoalignment of the reads was performed using the *kallisto* program [33]. Default parameters were used and 50 bootstraps were set up to estimate the technical error in the downstream analysis. Reads were pseudoaligned to the wheat transcriptome (The International Wheat Genome Sequencing Consortium–IWGSC RefSeq v2.0 assembly) [34]. 

Nonaligned reads from the second experiment were then pseudoaligned against the *Fusarium graminearum* PH1 transcriptome. Unmapped reads from both experiments were then pseudoaligned to the transcriptome of three closely related *Penicillium* spp. for reference (*P. chrysogenum* str. P2niaD18, *P. expansum* str. CMP-1 and *P. griseofulvum* str. PG3.). All the genomes are available at http://ensemblgenomes.org (accessed on 6 March 2022). 

### 2.4. Gene Ontology Analysis

In order to identify clusters of transcripts with similar expression patterns, we used the self-organizing map (SOM) competitive learning algorithm from the *kohonen* package in R [35,36]. Transcripts with fewer than 0.5 tpm were discarded. The mean of each time point and condition was calculated. These values were normalised to 0 with standard deviation from −1 to 1. SOMs were laid down using 100 nodes. Clusters were subdivided using hierarchical clustering. The final number of clusters was decided based on the point were within-group variation reached a low plateau [37]. Individual clusters of genes were then analysed for the enrichment of gene ontology terms (GO terms) with a 0.05 cut-off for false discovery rate using the *goseq* Bioconductor package [38]. 

### 2.5. Differential Gene Expression Analysis of Specific Genes Involved in Fusarium–Wheat Interaction

Clusters of genes, such as WRKY transcription factors, pathogenesis-related proteins (PR), antifungal proteins and specialised metabolites, reported to be active during *Fusarium*–wheat interactions were examined. Similarly, genes naturally expressed during anthesis and grain filling stage in wheat were inspected. Furthermore, *Fusarium* genes involved in trichothecene production such as the *TRI* cluster and *CLM1* were also analysed.

Differentially expressed genes were calculated at transcript level using a Wald test on ‘beta’ coefficients (analogous to Fold-change value) using the *sleuth* tool [39]. This value uses the bootstraps values from *kallisto* to estimate the ‘inferential variance’ or technical error. *p*-values were aggregated to determine gene-level results and then adjusted for multiple comparisons using a 0.1 cut-off for false discovery rate and transformed into *q*-values.

### 2.6. Multi-Mycotoxin and Specialised Metabolite Analysis

Wheat spikes from experiment 2.2 were used to quantify mycotoxins and specialised metabolites. The plant material was freeze dried for 48 h and homogenized using ceramic beads (10 mm) in a mechanical shaker (SO-40a, Fluid Management Europe B.V.) using a 30 s programme five times. For each sample, 3–5 full spikes were pooled to achieve 1 g of dried material and four replications per treatment were prepared. Five grams of a previously ground sample was extracted for 90 min using 20 mL of acetonitrile/water/acetic acid 79/20/1 (*v*/*v*/*v*). Particles in the raw extracts were removed by sedimentation by keeping the sample suspensions in an upright position for 10 min. An aliquot of the clear extract was diluted 1 + 1 with acetonitrile/water/acetic acid 20/79/1 and, subsequently, analysed (injection volume: 5 μL).

The samples were analysed with a multi-mycotoxin method based on LC-MS/MS analyses using an Agilent 1290 HPLC coupled to an Applied Biosystems 5500 QTrap mass spectrometer. Over 500 different compounds, including major mycotoxins, were measured simultaneously: trichothecenes, zearalenone, aflatoxin, fuminosins, T-2 toxin, depsipeptides, bacterial metabolites, *Fusarium* metabolites and other unspecific metabolites. This was performed at the Department for Agrobiotechnology in Tulln, part of the University of Natural Resources and Life Sciences (BOKU) in Austria, a research partner of BIOMIN^®^ according to the methods described previously [40]. Mycotoxin concentration data were analysed using a linear model. Comparisons between treatments were considered significant at *p* < 0.05.

## 3. Results

### 3.1. Penicillium olsonii ML37 Activates Defence Mechanisms at Heading-Stage Wheat Spikes 

*P. olsonii* ML37 inoculation changed the transcriptomic profile of wheat spikes at 48 h after inoculation. The SOM clustering and gene ontology analysis detected a group of transcripts highly expressed in ML37-treated plants. This cluster was significantly enriched with genes involved in defence responses, specifically defence to fungi, as well as stress-related and respiratory burst (Figure 1A).

The differential expression analysis showed that *P. olsonii* ML37 inoculation upregulated 105 genes and downregulated 48 genes (Appendix A Appendix A). Among the upregulated genes, several transcription factors involved in biotic and abiotic stress responses such as WIR1A, WRKY51, WRKY10 [41,42,43] were detected. Some key pathogenesis-related genes, such as PR-1 and PR-2, showed increase expression, although they were not statistically significantly upregulated (Figure 2). While some others such as PR1-16 and chitinase IV (Cht4) were significantly upregulated (Appendix A).

In addition, a key regulator of abscisic acid (ABA) catabolism, the ABA 8′-hydroxylase ABA8 gene [44], was significantly overexpressed.

Specific β-1,3-glucanase activity showed a significant increase at 48 h on *P. olsonii* ML37-treated spikes compared to the mock-treated ones, but the enzymatic activity declined and was not significant at 72 h (Appendix A). This confirmed that ML37 inoculation activates microbe-related defence responses in the plant after inoculation. 

Among the downregulated genes, there was a significant enrichment of GO terms related to cell proliferation, cell cycle, DNA regulation and the heterochromatin structure (Figure 1B). Several H2 and H3 histone genes were downregulated in ML37-treated plants. Notably, the largest part of transcripts in the samples (approximately 50%) remained unaffected by the treatments (Figure 1C). Among these, GO terms associated with photosynthesis and biological processes, such as translation, were over-represented. Several genes reported to be highly active during anthesis, such as α-amylase and ATP-dependent 6-phosphofruktinase [45,46] involved in carbohydrate transport, were not differentially expressed (Appendix A).

### 3.2. P. olsonii ML37-Treated Plants Show Higher Levels of Defence Responses during the First 24 h of FHB Attack

Wheat response to Fusarium infection varied drastically over time between ML37-treated and mock-treated plants. At 1 dpi, 293 genes were differentially expressed between treatments, while at 3 dpi, this number increased to 18,442 genes (Appendix A). A large cluster of genes significantly enriched with GO terms involved in response to chitin, cell death, pathogenesis, defence response pathways and others showed a higher expression levels in ML37-inoculated plants at 1 dpi and a steep decrease at 3 dpi (Figure 3A).

Within this cluster, key marker genes for the pathogenesis and activation of defence responses, such as PR-1-1, PR-1-9, PR-2, PR-4A [47,48] and others with antifungal activity such as chitinases, endochitinases and specialised metabolite production, were significantly overexpressed in ML37-treated plants compared to mock-treated plants at 1 dpi (Appendix A). These same genes showed the opposite trend at 3 dpi, as mock-treated plants showed a higher expression than the ML37-treated ones (Figure 4).

ML37-treated wheat spikes under FHB infection showed an increased specific β-1,3-glucanase activity as early as 24 dpi after pathogen inoculation compared to the mock-treated ones. This effect was greater at 48 h and declined at 72 dpi (Appendix A). 

An equally large second cluster of transcripts showed lower levels of expression at 1 dpi in ML37-treated plants and a higher expression at 3 dpi, while remaining constantly expressed in mock-treated plants across both time points (Figure 3B). This cluster was significantly enriched with GO terms associated with thylakoid, chloroplast relocation and photosynthesis, as well as floral development, cell division, starch metabolic process and postembryonic development. 

A third, smaller cluster with constantly high levels of expression at both 1 and 3 dpi in the ML37-treated plants while also being expressed at low levels in the mock-treated plants was detected. This was significantly enriched with GO terms involved in cell wall formation and maintenance such as pectin esterase activity (Figure 3C).

### 3.3. Fusarium graminearum Shows a Reduced Metabolism during P. olsonii ML37-Mediated Biocontrol

Only 2–15% of the unmapped reads were successfully aligned to the Fusarium graminearum transcriptome. Such low read quantities impeded a clearer visualization of the transcriptomic state of the pathogen during infection using Gene Ontology tools. Still, SOM clustering identified several groups of genes that showed reduced expression patterns on ML37-treated plants compared to the mock-treated ones at both time points (Appendix A). None of these clusters were significantly enriched with any GO terms after *p*-value adjustment, but the most abundant were related to ribosomal function and mRNA translation.

On the other hand, 112 genes were detected as differentially expressed in F. graminearum on ML37-treated spikes compared to mock-treated spikes at 1 dpi. This number increased to 228 at 3 dpi (Appendix A). Interestingly, some of the genes significantly overexpressed in the ML37-treated spikes were involved in stress-related responses. Among these were ABC transporter multidrug resistance protein 1, cytochrome P45O monooxygenase and antifungal defensin (Appendix A).

We further inspected the expression of several fungal genes involved in trichothecene biosynthesis, namely, TRI-6, TRI-5 and TRI-12 [49]. These genes showed low expression levels at 3 dpi, but they were not significantly affected by ML37 inoculation at this early stage. Interestingly, CLM1, a gene involved in the synthesis of culmorin-like metabolites, was significantly repressed in ML37-treated plants at 3 dpi (Figure 5).

Spikes were assessed for FHB severity at 7 dpi and a significant reduction in infection was observed on the ML37-treated plants (Figure 6). The LC-MS/MS analysis of over 500 microbial specialised metabolites in full spikes at 7 dpi revealed lower concentrations of trichothecenes in ML37-treated spikes compared to mock-treated spikes, including statistically significant reductions in 15-acetyl-deoxynivalenol and culmorin (Table 1).

### 3.4. Expression of Penicillium Genes Not Detected, but Fungal Metabolites Were Detected on Treated Spikes

Unmapped reads from experiment two were pseudoaligned to the genomes of three closely related *Penicillium* spp., i.e., *P. chrysogenum*, *P. expansum* and *P. griseofulvum* [50]. On average, only 0.5 of the reads used aligned against these genomes. Moreover, there were no differences in the number of reads mapped between the ML37-treated and nontreated plants. Thus, we concluded that the detection of ML37 transcripts was unsuccessful.

The multi-mycotoxin method detected significantly higher concentrations of the fungal metabolites N-benzoyl-phenylalanine and asperphenamate on the ML37-treated spikes compared to mock-treated spikes after Fusarium infection (Table 1). These molecules were not detected on the untreated controls.

## 4. Discussion

*Penicillium olsonii* ML37 has been identified as a new potential biocontrol agent against wheat pathogens [30,31]. Here, we present the transcriptomic response of wheat spikes after inoculation with *P. olsonii* ML37 alone and during *Fusarium* infection. Our results highlighted the importance of timing behind the biological control of FHB as ML37-treated plants showed a reduced disease severity and mycotoxin accumulation.

The transcriptomic data showed that *P. olsonii* ML37 spores that landed on the wheat spikes were recognized by the plant. Hence, the transcription factors *WRKY10, WIR1A* and *WRKY51*, which are associated with the response to stress [51], were upregulated by ML37. These genes are also highly responsive to the exogenous application of purified chitin in Arabidopsis [52]. Moreover, the expression of *WRKY10* and *WIR1A* has been associated with the deployment of basal resistance in rice [53] as well as the downstream activation of pathogen responsive genes in the cell wall [42]. As far as we are aware, this was the first time WRKY gene induction was seen following colonisation with an endophytic fungus.

The recognition of *P. olsonii* ML37 triggered plant defence mechanisms that led to the local expression of PR-proteins and a significant increase in β-1,3-glucanase activity in treated spikes. This is a common response after interactions with beneficial organisms. It is part of a basal defence mechanism called induced resistance [54]. Similar responses have been observed in other crops after inoculation with different microorganisms, such as *Trichoderma* in maize [55,56] and *Bacillus* spp. in soybean [57], and they have also been reported during interactions between plants and endophytic fungi [58,59].

The basal defence response to *P. olsonii* ML37 caused an active reprogramming of metabolic processes in the wheat plants. Histone-related genes and chromatin configuration genes were significantly downregulated after ML37 inoculation. Epigenetic regulation has been suggested to play a key role in the deployment of defence responses in general and, specifically, in the onset of defence responses elicited by beneficial microbes [60,61]. Simultaneously, this activation reduced photosynthetic activity on wheat spikes. This is a common consequence of plant defence activation, and its effects on plant fitness have been discussed previously [62]. However, it has been shown that changes in photosynthesis and growth-related genes do not change the expression of normal anthesis developmental processes [45,46].

*Fusarium*-inoculated spikes that were pretreated with *P. olsonii* ML37 spores showed a higher expression of genes involved in pathogen defence, the oxidative burst, PR proteins, chitinases, glycosyltransferases as well as a higher specific β-1,3-glucanase activity at 1 dpi than in the mock-treated spikes. Interestingly, several PR-genes (*PR1-1, PR1-9, PR1-16, PR4A and PR4B)* showed a greater expression in the mock-treated spikes compared to the ML-37-treated ones at 3 dpi, suggesting a delayed response for these key defence genes. By having a higher expression of these genes at 24 h, the pretreated plants were able to minimize the initial infection, while the mock-treated plants showed a relatively lower expression at this point. This pattern was reversed at 3 dpi, where the mock-treated plants showed a higher expression of these genes. This suggested a stronger defence response to a high infection level of *Fusarium*. This showed that activating defence responses during the first 24 h after pathogen arrival is the key driver of the *P. olsonii* ML37-mediated effect. Thus, our data strongly suggest that the timing of the defence response deployment is a key factor in the mode of action of ML37-mediated biocontrol, as the early activation of defence mechanisms was associated with FHB suppression.

The rapid activation and deployment of defence responses has been associated with FHB resistance in several cultivars. A greater activation of the defence genes *PR-1* and *PR-2* at 6-12 hpi was observed in the tolerant cultivar Sumai 3 [11]. In wheat, these genes have been reported to be highly responsive to fungal pathogens [11,63,64].

We showed that specific β-1,3-glucanase activity peaked at 48 hpi, but decreased at 72 hpi in the ML37-treated compared to the mock-treated plants. These proteins are produced after both the SA and JA pathway activation, representing the activity of several different enzymes. They inhibit pathogen growth by hydrolysing the main chain of β-1,3-glucans in fungal cell walls [65,66]. They have been observed in transcriptomic and proteomic studies in FHB-tolerant cultivars [12,67].

We observed a high expression of glycosyltransferase and sulfotransferase proteins in ML37-treated compared to mock-treated spikes. These enzymes use glucosinolates, brassinosteroids, jasmonates and flavonoids as substrates to generate specialised metabolites. These types of enzymes, especially uridinediphosphate glycosyltransferases and glutathione S-transferases, have been associated with major FHB-disease resistance QTLs [68], presumably by the detoxification of DON into less toxic forms such as D3G [15]. In our study, the simultaneous activation of antifungal enzymes and specialised metabolite production was associated with a reduced translation and ribosomal activity in the pathogen, and likely to be the cause of the reduced FHB symptoms in ML37-treated plants.

The expression of three key trichothecene pathway genes during the first 72 h was not affected by the *P. olsonii* ML37 treatment. However, there was a significant reduction in the accumulation of trichothecenes and other *Fusarium* metabolites in whole wheat spikes after 7 days. The sesquiterpenoid mycotoxins, 15-Acetyl-DON and culmorin, were significantly reduced in ML37-treated plants. 15-Acetyl-DON is the acetylated derivate of DON and possesses equally regulatory importance, whereas culmorin is a co-occurring *Fusarium* metabolite, which has been suggested to be an “emerging mycotoxin” [69]. It enhances DON accumulation by inhibiting DON detoxification via glycosylation in plants [70]. Interestingly, Wipfler et al. [71] reported that culmorin biosynthesis genes were more highly to be expressed than the *TRI5* trichothecene biosynthetic gene during the first week after infection. This suggests that ML37-mediated defence activation in wheat spikes prevents culmorin production during the first 72 h after infection, consequently, reducing trichothecene production and the onset of the necrotrophic phase of FHB. This was further validated by the RNA-seq data showing the repression of the gene *CLM1* involved in the synthesis of culmorin.

Previous studies have suggested an indirect mode of action of some biocontrol agents. For example, Jochum et al. [72] showed that *Lysobacter enzymogenes* C3 heat-inactivated cultures reduced FHB to the same extent as fresh cultures when sprayed 8 days before *F. graminearum* inoculation. Other studies have used plant hormone accumulation as an indication of an increased defence activation. Palazzini et al. [23] suggested the phytohormone induction of SA and JA by *Bacillus velezensis* RC-218 when co-inoculated with *Fusarium graminearum*, but this effect was not consistently higher than the *F. graminearum* control during the first 72 h, while the bacteria alone showed no differences compared to the mock (water) control. Similarly, Kemp et al. [29] showed that spikes pre-inoculated with the endophytic yeast *Sarocladium zeae* showed higher levels of both SA and JA during FHB compared to the *F. graminearum*-only control. However, this effect was only significant at 1 h after pathogen inoculation for SA and up to 8 h for JA. In neither of these studies was there direct evidence of defence mechanisms known to inhibit *Fusarium* spp, and/or their dynamics before and after pathogen inoculation. Our study was the first to fully describe the effective biological control of FHB through plant defence activation.

*P. olsonii* ML37 has shown significant biocontrol effects against two major pathogens in wheat, namely, *Z. tritici* and *F. graminearum* [30,31]. In this study, we were unable to detect *P. olsonii gene* expression. However, we were able to detect high amounts of the linear amino acid ester asperphenamate and one of its components, N-benzoylphenylalanine, only in the ML37-treated spikes, further confirming an active role of ML37. These molecules are produced by fungi such as *Aspergillus* and *Penicillium* and they are particularly common among endophytic fungi [73]. They are biologically active and have been linked to several chemical and pharmaceutical applications [74], as well as displaying antimicrobial properties [75]. However, *P. olsonii* ML37 does not inhibit the growth of *Z. tritici* nor *F. graminearum* in vitro. It is possible that asperphenamate and N-benzoylphenylalanine are only produced in planta. In fact, asperphenamate has been detected in diverse groups of unrelated plants, which could suggest a common generalist fungal endophyte origin [74,76].

Multiple and simultaneous modes of action are a common trait of biocontrol agents [77], especially in fungal endophytes [78]. For example, the biocontrol agent *Clonostachys rosea* is a mycoparasite [79] with the ability to detoxify the *Fusarium* mycotoxin zearalenone [80]. Similarly, mycoparasitism and competition were shown in biological control by *Trichoderma gamsii* T6085 against *Fusarium* spp. [81]. Although it is likely that ML37 possesses several modes of action, this remains to be verified using targeted experiments.

Recently, it was shown how *fhb7*, a gene that confers the resistance gene against Fusarium head blight in wheat, has a fungal endophyte origin and was transmitted to wild relatives of wheat via a horizontal gene transfer from *Epichloë* fungi [82]. This highlights the degree of coevolution between fungal endophytes and their host plants. Indeed, naturally occurring fungal endophytes have been observed to be associated with healthy wheat spikes and negatively correlated with *Fusarium* abundance [83]. Here, we showed how *P. olsonii* ML37 increased wheat tolerance to biotic stress caused by *F. graminearum*. This capacity has the potential to be used as a novel biocontrol strategy against FHB in wheat production.

## Figures and Tables

**Figure 1 jof-08-00345-f001:**
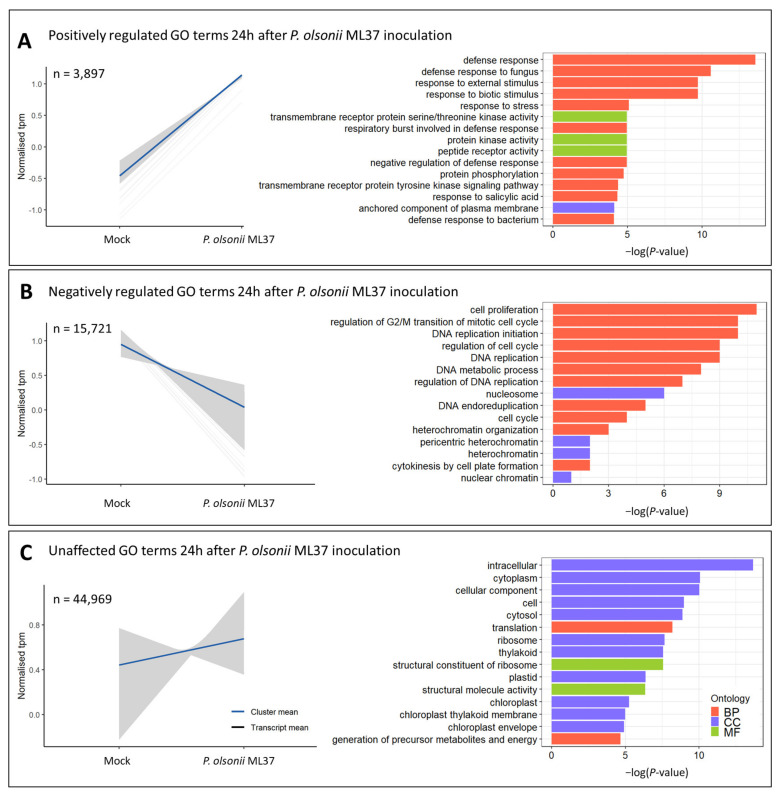
Transcriptome changes induced by *P. olsonii* ML37 in wheat spikes at 48 h after inoculation. (**A**) (left) Normalised transcripts per million (tpm) of a cluster of genes with positive regulation after ML37 inoculation. Grey lines represent mean expression of transcripts, blue line represents total cluster mean. (**A**) (right) Significantly enriched gene ontology terms within each cluster. Bar plots represent −log(*p*-value) of each GO term. (**B**) (left) Normalised tpm of a cluster of genes with negative regulation after ML37 inoculation. (**B**) (right) Significantly enriched GO terms. (**C**) (left) Normalised tpm of a cluster of genes unaffected by ML37 inoculation. (**C**) (right) Significantly enriched GO terms. Gene ontology conventions—BP: biological process; CC: cell component; MF: molecular function.

**Figure 2 jof-08-00345-f002:**
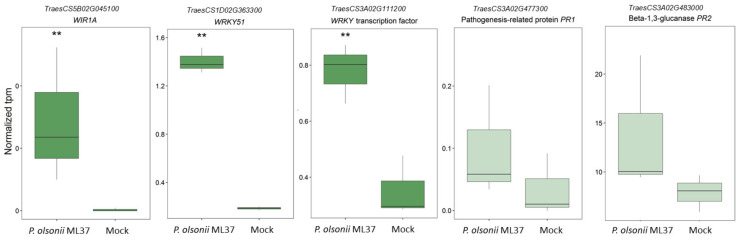
Expression of three WRKY transcription factors and two pathogenesis-related genes in wheat spikes after inoculation with *P. olsonii* ML37 alone before anthesis. Boxplot shows range and median of mean tpm for each gene. Statistically significant differences (*q* < 0.01 using Walt test) are denoted as ‘**’. Deep green colour denotes differentially expressed genes. Light green colour denotes non-differentially expressed genes.

**Figure 3 jof-08-00345-f003:**
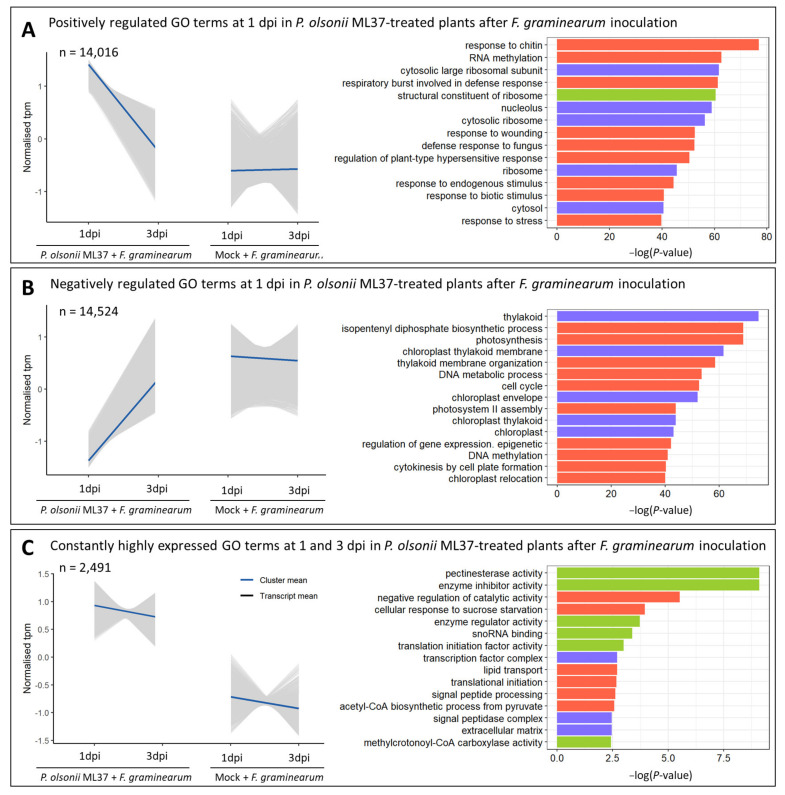
Transcriptome changes during *P. olsonii* ML37-mediated biocontrol of FHB at 1 and 3 dpi. (**A**) (left) Normalised transcripts per million (tpm) of a cluster of genes with positive regulation in ML37-treated plants after *F. graminearum* inoculation. Grey lines represent mean expression of transcripts, blue line represents total cluster mean. (**A**) (right) Significantly enriched gene ontology terms within each cluster. Bar plots represent –log(P-value) of each GO term. (**B**) (left) Normalised tpm of a cluster of genes with negative regulation in ML37-treated plants after pathogen inoculation. (**B**) (right) Significantly enriched GO terms. (**C**) (left) Normalised tpm of a cluster of genes highly expressed at both time points in ML37-treated plants. (**C**) (right) Significantly enriched GO terms. Gene ontology conventions—BP: biological process; CC: cell component; MF: molecular function.

**Figure 4 jof-08-00345-f004:**
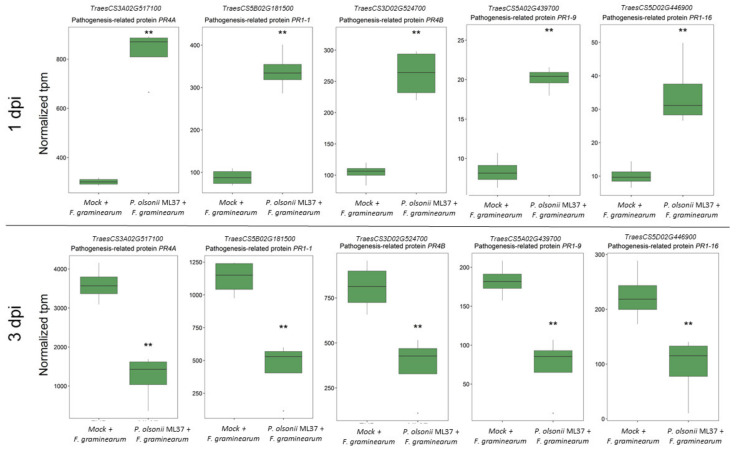
Expression of five different Pathogenesis-related (PR) genes in *P. olsonii* ML37 and mock-treated spikes at 1 and 3 dpi with *F. graminearum*. Boxplot shows range and median of mean tpm for each gene. Statistically significant differences (*q* < 0.01 using Walt test) are denoted as ‘**’.

**Figure 5 jof-08-00345-f005:**
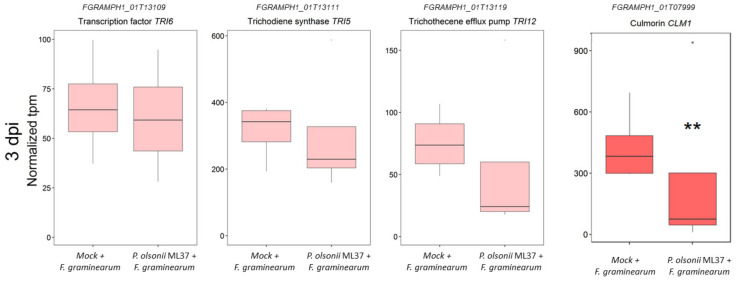
Expression of three genes from the TRI cluster and CLM1 involved in deoxynivalenol and culmorin production, respectively, on endophyte-treated spikes and FHB control at 3 dpi. Boxplot shows range and median of mean tpm for each gene. Light red colour denotes nondifferentially expressed genes. Boxplot shows range and median of mean tpm for each gene. Statistically significant differences (*q* < 0.01 using Walt test) are denoted as ‘**’.

**Figure 6 jof-08-00345-f006:**
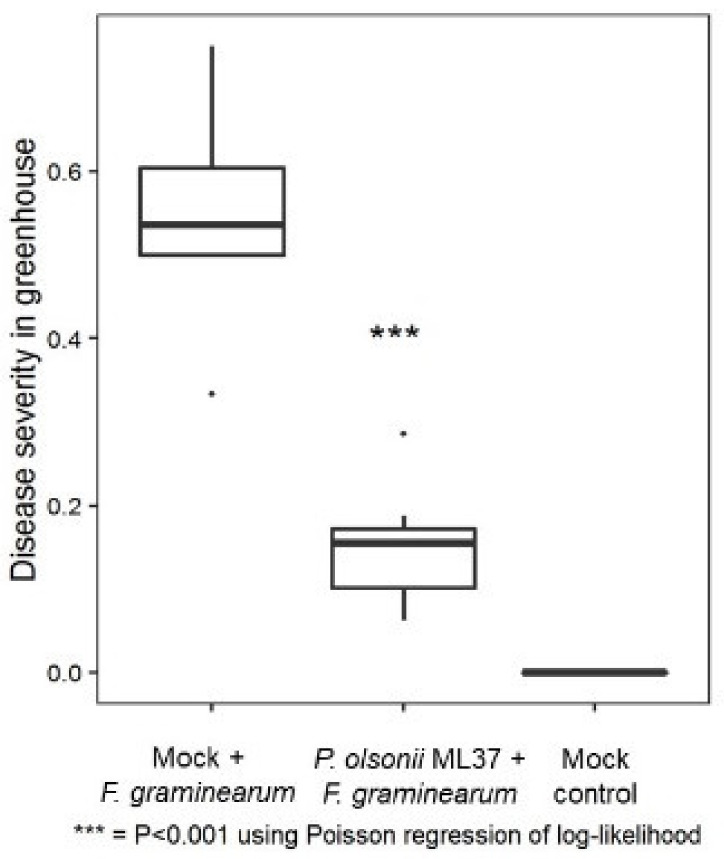
Effect of *Penicillium olsonii* ML37 on FHB severity at 5 dpi with *F. graminearum* under greenhouse conditions. Boxplot shows range and median severity for each treatment. Points below and up boxes are outlier data points. Statistically significant differences at *p* < 0.001 compared to FHB control are denoted as ‘***’.

**Table 1 jof-08-00345-t001:** HPLC-ESI-MS/MS-based multi-mycotoxin analysis of wheat spikes treated with *P. olsonii* ML37 at 7 dpi with *Fusarium graminearum.*

		Concentration ppb (µg/kg)	
Type	Molecule	Mock	*P. olsonii* ML37
Trichothecenes	Deoxynivalenol	106,602 ^a^	84,390 ^a^
	15-Acetyildeoxynivalenol	7618 ^a^	4784 ^b^
	DON-3-glucoside	10,511 ^a^	7929 ^a^
	Nivalenol	115 ^a^	89 ^a^
*Fusarium*-specific metabolites	Culmorin	183,035 ^a^	133,627 ^b^
	15-Hydroxyculmorin	64,928 ^a^	54,355 ^a^
	15-Hydroxyculmoron	8188 ^a^	6133 ^a^
	5-hydroxyculmorin	11,725 ^a^	9484 ^a^
Unspecific metabolites	N-Benzoyl-Phenylalanine	0 ^b^	34 ^a^
	Asperphenamate	35 ^b^	1055 ^a^
	Altersetin	8 ^a^	0 ^b^

^a^ and ^b^ denote statistical differences (*p*-value < 0.05) using Fisher’s exact test.

## Data Availability

The sequencing data that support the findings of this study are available in the NCBI repository under SRA submission SUB11165978. Other experimental data are available upon request.

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
