# Peer review of "The Fungal Endophyte Penicillium olsonii ML37 Reduces Fusarium Head Blight by Local Induced Resistance in Wheat Spikes"

_jof, 2022, doi:10.3390/jof8040345_

Round 1
Reviewer 1 Report
The article “The fungal endophyte Penicillium olsonii ML37 reduces Fusarium head blight by local induced resistance in wheat spikes” provide an interesting endophyte Penicillium olsonii that inhibit pathogen infection by induced resistance. Endophyte fungi are good biocontrol agents. I suggested this article can be accepted for publication. While some questions needed be considered before publication.
- At line 132, each spike was inoculated with how much volume of F. graminearum WC-091-7 macroconidial suspension (105spores/ml)?
- Pathogenesis related gene PR-2 (β-1,3-glucanase), was not statistically significantly upregulated. But β-1,3-glucanase activity showed a significant increase at 48 h on P. olsonii ML37 treated spikes compared to mock-treated. These two results were not unanimous. How do you explain this result?
- In figure 4, Expression of five different Pathogenesis-related (PR) genes in P. olsonii ML37-treated spikes increased at 1 dpi but decreased at 3 dpi with F. graminearum. Please explain the opposite result at 1 and 3 dpi.
Author Response
The article “The fungal endophyte Penicillium olsonii ML37 reduces Fusarium head blight by local induced resistance in wheat spikes” provide an interesting endophyte Penicillium olsonii that inhibit pathogen infection by induced resistance. Endophyte fungi are good biocontrol agents. I suggested this article can be accepted for publication. While some questions needed be considered before publication.
<< Thank you for the kind words and constructive suggestions
- At line 132, each spike was inoculated with how much volume of F. graminearum WC-091-7 macroconidial suspension (105spores/ml)?
There is unfortunately no unequivocal answer since spray inoculation was used, and the spikes were sprayed until run-off, which is the traditional way inoculation is described. Roughly, 2 ml was used per spike. This is included at line 114 and line 133-134.
2. Pathogenesis related gene PR-2 (β-1,3-glucanase), was not statistically significantly upregulated. But β-1,3-glucanase activity showed a significant increase at 48 h on P. olsonii ML37 treated spikes compared to mock-treated. These two results were not unanimous. How do you explain this result?
There is rarely a correlation between mRNA levels and enzyme activity. In addition, the enzyme activity represents the activity of several enzymes, whereas the PR2 transcript was only one at least six copies of that gene in the wheat genome. We have changed the text as follows (lines 424-429):
We showed that specific β-1,3-glucanase activity peaked at 48 hpi, but decreased at 72 hpi in the ML37-treated compared to the mock-treated plants. These proteins are produced after both SA and JA pathway activation, representing the activity of several different enzymes. They inhibit pathogen growth by hydrolysing the main chain of β-1,3-glucans in fungal cell walls [65,66]. They have been observed in transcriptomic and proteomic studies in FHB tolerant cultivars [12,67].
3. In figure 4, Expression of five different Pathogenesis-related (PR) genes in P. olsonii ML37-treated spikes increased at 1 dpi but decreased at 3 dpi with F. graminearum. Please explain the opposite result at 1 and 3 dpi.
We have strengthened and modified this discussion in the paragraph in lines 405-419:
Fusarium inoculated spikes that were pre-treated with P. olsonii ML37 spores showed higher expression of genes involved in pathogen defence, the oxidative burst, PR proteins, chitinases, glycosyltransferases as well as higher specific β-1,3-glucanase activity at 1 dpi than in mock-treated spikes. Interestingly, several PR-genes (PR1-1, PR1-9, PR1-16, PR4A and PR4B) showed greater expression in the mock-treated spikes compared to the ML-37 treated at 3 dpi, suggesting a delayed response for these key defence genes. By having higher expression of these genes at 24 h, the pre-treated plants are able to minimize the initial infection. While the mock treated plants showed relative lower expression at this point. This pattern is reversed at 3 dpi where mock-treated plants showed higher expression of these genes. This suggests a stronger defence response to a high infection level of Fusarium. This shows that activating defence responses during the first 24 h after pathogen arrival is the key driver of the P. olsonii ML37 mediated effect. Thus, our data strongly suggest that timing of defence response deployment is a key factor in the mode of action of ML37-mediated biocontrol, as early activation of defence mechanisms was associated with FHB suppression.
Reviewer 2 Report
The manuscript describes P. olsonii M37 as a biocontrol agent against Fusarium head blight (FHB). Alternative controls to the use of chemical fungides are getting attention worldwide since we have concerns about human health, resistant pathogens, and enviromental contamination. So, the subject of the manuscript is very important at the present time.
In order to understand how P. olsonii M37 acts in the endophytic-host-pathogen interaction, the authors use two experimental approaches: genome (RNA-seq) and metabolomics (LC-MS/MS) analysis, which are not much innovative, but were employed correctly.
The results are not in doubt, but my most substantial point is regarding to the LC-MS/MS analysis and metabolite identification, which is poorly desribed in the manuscript and can be improved.
- How the metabolite extraction were perfomed of the dried plant material?
- I know that the authors described the reference used for the LC-MS/MS analysis (40). But informations are missing. By example, how the samples were prepared after the extraction (concentration mg/mL)? Were the samples filtered? Please, provide more details.
- Informations about the metabolite indentification need to be provide in the supplemental material. Chromatograms and mass spectra should be shown. Information about retention time, m/z values, m/z errors, fragmentation patterns (MS/MS) and comparision with standard molecules will make the identifications more solid.
The work can be published after the corrections.
Author Response
The manuscript describes P. olsonii M37 as a biocontrol agent against Fusarium head blight (FHB). Alternative controls to the use of chemical fungides are getting attention worldwide since we have concerns about human health, resistant pathogens, and environmental contamination. So, the subject of the manuscript is very important at the present time.
In order to understand how P. olsonii M37 acts in the endophytic-host-pathogen interaction, the authors use two experimental approaches: genome (RNA-seq) and metabolomics (LC-MS/MS) analysis, which are not much innovative, but were employed correctly.
The results are not in doubt, but my most substantial point is regarding to the LC-MS/MS analysis and metabolite identification, which is poorly desribed in the manuscript and can be improved.
<<Thank you for the kind words and constructive suggestions
1. How the metabolite extraction were perfomed of the dried plant material?
This is described by the Austrian contractor in reference 40. Michael Sulyok tells us that they have found that all filter materials may adsorb traces of some analytes (enniatins, beauvericin and fumonisins are particularly critical in that aspect). Particles in the raw extracts are removed by sedimentation by keeping the samples suspension in an upright position for 10 min. An aliquot of the clear extract is diluted 1+1.
The background information is available Open Access as Electronic Supporting material Table 2.
https://doi.org/10.1007/s00216-020-02489-9
We have therefore added the following in the description in lines 206-215):
Multi mycotoxin and specialised metabolite analysis
Wheat spikes from experiment 2.2 were used to quantify mycotoxins and specialised metabolites. The plant material was freeze dried for 48 h and homogenized using ceramic beads (10 mm) in a mechanical shaker (SO-40a, Fluid Management Europe B.V.) using a 30-s programme five times. For each sample, 3-5 full spikes were pooled to achieve 1 g of dried material and four replications per treatment were prepared. Five grams of previously ground sample was extracted for 90 min using 20 ml acetonitrile/water/acetic acid 79/20/1 (v/v/v). Particles in the raw extracts were removed by sedimentation by keeping the sample suspensions in an upright position for 10 min. An aliquot of the clear extract was diluted 1+1 with acetonitrile/water/acetic acid 20/79/1 and subsequently analysed (injection volume: 5 μl).
2. I know that the authors described the reference used for the LC-MS/MS analysis (40). But informations are missing. By example, how the samples were prepared after the extraction (concentration mg/mL)? Were the samples filtered? Please, provide more details.
Please see the reply above.
3. Informations about the metabolite indentification need to be provide in the supplemental material. Chromatograms and mass spectra should be shown. Information about retention time, m/z values, m/z errors, fragmentation patterns (MS/MS) and comparision with standard molecules will make the identifications more solid.
The requested methodology information is available Open Access as Electronic Supporting material Table 2 in reference 40 (Sulyok et al. 2020), https://doi.org/10.1007/s00216-020-02489-9. We would like to refer the reader to the publication rather than explaining all details in the paper.
The data acquisition mode used for quantitative analysis (i.e. the Scheduled Multiple Reaction Monitoring Mode) does not include acquisition of full mass spectra. Instead, official criteria for unambiguous confirmation of the identity of the analytes have been set based on retention time and intensity ratio of the two acquired MS/MS transitions in comparison to an authentic standard
https://ec.europa.eu/food/system/files/2017-05/cs_contaminants_sampling_guid-doc-ident-mycotoxins.pdf
In that aspect, the method complies to the 30% relative set for the intensity ratio and uses a more strict in house criterion for LC retention time (tolerance of 0.03 min instead of 0.2% rel.).